# New Insights on the Effect of TNF Alpha Blockade by Gene Silencing in Noise-Induced Hearing Loss

**DOI:** 10.3390/ijms21082692

**Published:** 2020-04-13

**Authors:** Janaína C. Rodrigues, André L. L. Bachi, Gleiciele A. V. Silva, Marcelo Rossi, Jonatas B. do Amaral, Karina Lezirovitz, Rubens de Brito

**Affiliations:** 1Clinical Hospital, Department of Otorhinolaryngology-Head and Neck Surgery, School of Medicine, University of São Paulo (USP), São Paulo 05403-000, Brazil; lezi.karina@gmail.com (K.L.); rbritoneto@gmail.com (R.d.B.); 2Laboratory of Otolaryngology (LIM32), School of Medicine, University of São Paulo (USP), São Paulo 05403-000, Brazil; gleiciele.alice@outlook.com; 3ENT Research Lab. Department of Otorhinolaryngology-Head and Neck Surgery, Federal University of São Paulo. (UNIFESP), São Paulo-SP 04039-032, Brazil; allbachi77@gmail.com (A.L.L.B.); mrossi@dim.fm.usp.br (M.R.); amaraljb@gmail.com (J.B.d.A.); 4Brazilian Institute of Teaching and Research in Pulmonary and Exercise Immunology (IBEPIPE), São José dos Campos 12245-520, Brazil; 5Post-graduation Program in Health Sciences, Santo Amaro University (UNISA), São Paulo 04829-300, Brazil

**Keywords:** TNFα metabolic pathway, apoptosis, auditory brain response, cochlea, electrophysiological threshold, in vivo siRNA administration, synaptopathy

## Abstract

Noise exposure represents the second most common cause of acquired sensorineural hearing loss and we observed that tumor necrosis factor α (TNFα) was involved in this context. The effect of *Tnfα* gene silencing on the expression profile related to the TNFα metabolic pathway in an experimental model of noise-induced hearing loss had not previously been studied. Methods: Single ears of Wistar rats were pretreated with *Tnfα* small interfering RNA (siRNA) by trans-tympanic administration 24 h before they were exposed to white noise (120 dBSPL for three hours). After 24 h of noise exposure, we analyzed the electrophysiological threshold and the amplitude of waves I, II, III, and IV in the auditory brain response click. In addition, qRT-PCR was performed to evaluate the TNFα metabolic pathway in the ears submitted or not to gene silencing. Results: Preservation of the electrophysiological threshold and the amplitude of waves was observed in the ears submitted to gene silencing compared to the ears not treated. Increased anti-apoptotic gene expression and decreased pro-apoptotic gene expression were found in the treated ears. Conclusion: Our results allow us to suggest that the blockade of TNFα by gene silencing was useful to prevent noise-induced hearing loss.

## 1. Introduction

Exposure to noise is one of the most common causes of acquired sensorineural hearing loss. It is threatening to public health due to the association of occupational noise exposure with recreational noise exposure in loud concerts, sporting events, and the increasing use of portable music players. The World Health Organization estimates that 1.1 billion young people are at risk for noise-induced hearing loss, mainly due to recreational exposure. Furthermore, hearing loss and tinnitus are an important disability claim for many people [1].

In order to guarantee a correct hearing loss diagnosis, it is widely accepted that the auditory brainstem response (ABR) represents a powerful clinical tool applied during a hearing screening, threshold estimation, and site-of-lesion testing [2]. Auditory brainstem response (ABR) involves short-latency evoked neural potentials, recorded from dermal or subdermal electrodes, that emerge as a series of six to seven vertex positive waves with latencies <10 milliseconds (ms) [3]. With transient stimuli (e.g., clicks or tone bursts), ABRs are commonly generated resulting in different waveforms [2] that are used to determine the occurrence or not of hearing loss and also its clinical follow-up.

In relation to noise-induced hearing loss, studies have demonstrated that apoptosis, necrosis, and/or necroptosis can occur in response to mechanical and/or metabolic damage and can lead to the degeneration of cochlear structures, such as hair cells, support cells, and afferent fibers of the cochlear nerve [4]. Corroborating this fact, the expression of pro- and anti-apoptotic genes in the peripheral auditory system (i.e., the cochlea) changes after noise exposure [5,6].

Furthermore, studies have demonstrated that inflammatory responses may also be involved in cochlear noise damage. It was found not only that the cytokines levels were increased after noise exposure but also that the blockade of certain cytokines, such as tumor necrosis factor α (TNFα), was able to preserve hearing [1,7,8]. Tumor necrosis factor-α is a classical pro-inflammatory cytokine involved in many different inflammatory and immune processes, as well as in the induction of apoptosis, differentiation, and cell recruitment. This multipotent cytokine can activate transcriptional pathways involved in the oxidative stress, inflammation, and apoptosis pathways, which led to cell degeneration [9,10,11].

It has been reported that TNFα is secreted by spiral ligament fibroblasts, outer ciliary cells, and support cells into the cochlea after noise-exposure and that interaction with the receptor TNFR1 (tumor necrosis factor receptor-1) increased the production of reactive oxygen species (ROS) and activated caspases in spiral ligaments [12,13,14]. Higher ROS levels induced the up-regulation of the inflammation cascade, which lead to the initiation of the extrinsic cell death pathway in auditory hair cells. In addition, TNFα can also initiate a MAPK (mitogen-activated protein kinase)/JNK (c-Jun-N-terminal kinase) signal cascade leading to phosphorylation of the transcription factor c-jun, which is associated with cell survival or apoptosis [15].

The use of different anti-TNFα agents, which can act by inhibition of the protein, such as infliximab, or by the decoy receptor, such as etanercept, can mitigate noise-induced hearing loss [11]. Based on this fact that the inhibition of TNFα was beneficial in noise-induced hearing loss, the blockade of TNFα by gene silencing could putatively provide relevant data to understand the real action of this cytokine in this context, due to gene silencing. This is also known as the use of small interfering RNA (siRNA) and is an innovative genetic regulatory mechanism that activates a sequence-specific RNA degradation process and allows the determination of a gene function under pathological conditions [16]. Therefore, in this study, we aimed to investigate the effect of *Tnfα* gene silencing on the expression profile related to the TNFα metabolic pathway in an experimental model of noise-induced hearing loss and also to determine the effect of TNFα blockade by gene silencing in the ABR click parameters.

## 2. Results

### 2.1. Tnf alpha siRNA Silencer Was Able to Reduce TNFα Expression

First, we conducted the in vitro silencer validation test by high-content screening fluorescence analysis using TNFα tagged in green. As shown in Figure 1A, when the scrambled siRNA was used, a high cell fluorescence was observed. However, the *Tnfα* siRNA administration (Figure 1B) led to significantly decreased fluorescence (*p* < 0.005) with a silencing rate of 96% (Figure 1C). Figure 1D shows the positive control of the delivery method used in this study. The positive control was provided by the commercial kit.

The in vivo silencing rate was 74.1% (*p* < 0.001) analyzed by real time qRT-PCR. This silencing ratio was calculated from the values obtained in the ears of rats submitted or not to *Tnfα* siRNA administration and after, to the noise exposure.

All the data related to the alteration of gene expression, and the fold-changes in the ears of rats obtained before and after noise exposure, as well as when the ears were submitted or not to *Tnfα* siRNA administration and after to the noise exposure are presented as Appendix A, respectively, and also Appendix A, respectively).

### 2.2. Tnf alpha siRNA Administration Elicited a Differential Gene Expression in Rats Submitted to Noise Exposure

The effect of TNFα gene silencing on the expression profile related to the TNFα metabolic pathway is shown in Figure 2, with the differential gene expression, carried out by qRT-PCR, between ears of rats submitted or not to TNFα blockade by gene silencing. All the genes analyzed were related to the TNFα metabolic pathway.

To improve the understanding of the characteristics of genes evaluated in this study, Table 1 shows the genes grouped in their respective families. In addition, this table also shows the fold changes of siRNA *Tnfα*/scramble and noise/no noise.

The genes that presented an Rq (fold change) of ±1.2657 were considered up and down-regulated by the *Tnfα* siRNA administration. Figure 3 shows the main apoptotic genes that were up and down-regulated in the ears submitted to TNFα blockade by gene silencing as compared to the gene expression found in the ears not submitted to gene silencing. 

### 2.3. TNF alpha Blockade by Gene Silence Was Able to Preserve the Hearing in Ears of Rats Submitted to Noise Exposure

In order to verify if the TNFα blockade by gene silencing could preserve the hearing of rats exposed to noise, we performed an analysis of the electrophysiological threshold shift in the ears of rats previously submitted (S) or not submitted (NS) to *Tnfα* siRNA administration. As showing in Figure 4, the ear submitted to *Tnfα* siRNA administration presented preservation of the electrophysiological threshold, whereas a significant hearing loss was found in the ear not submitted to *Tnfα* siRNA administration (scramble siRNA).

As shown in Figure 5, a higher mean electrophysiological threshold shift was found in the ears not submitted (NS) to the TNFα blockade by gene silencing (*n* = 20) as compared to the values observed both before the treatment (*p* < 0.001) and also to the ears submitted (S) to gene silencing (*n* = 20) (*p* <0.001).

In addition to the analysis of the electrophysiological threshold shift, the auditory brainstem response (ABR) was also carried out in order to demonstrate if the TNFα blockade by gene silencing had the capacity to preserve the hearing of rats after exposure to noise. As shown in Figure 6, the waves amplitude analysis at 90 dBSPL (left panel, A) in the ears submitted to TNFα blockade by gene silencing (S, *n* = 20) were higher (wave I, *p* < 0.001; wave II, *p* < 0.0001; wave III, *p* < 0.01; wave IV, *p* < 0.0001) as compared to the ears not submitted to TNFα blockade by gene silencing (NS, scramble, *n* = 20). In addition, the wave I amplitude at 80 dBSPL (*right panel*, B) was also increased (*p* < 0.05) in the ears submitted to TNFα blockade by gene silencing (S, *n* = 20) as compared to the ears not submitted to TNFα blockade by gene silencing (NS, scramble, *n* = 20).

## 3. Discussion

In this study, we were able to demonstrate, for the first time, that the blockade of TNFα through gene silencing maintained the electrophysiological threshold in the ears of the rats exposed to noise as compared to the harmful alterations observed in the ears not submitted to the gene silencing. In order to evaluate the mechanism involved in the maintenance of hearing in the ears submitted to the *Tnfα* blockade, we analyzed several genes of TNFα metabolism pathway and found a significant up-regulation of anti-apoptotic genes, as well as a downregulation of pro-apoptotic genes in the cochlea of ears submitted to TNFα blockade by gene silencing. 

Corroborating these observations, we observed that some pro-apoptotic genes (for instance *Casp 8*, *Tnfα*, *Tnfrsf1A*, *Tnfsf11b*, and *Fas* were positively-regulated only with noise exposure (as can be seen in Table 1-fold change in the situation “noise/no noise”). Furthermore, the blockage of TNFα by gene silencing was able to induce a significant alteration on these gene expressions leading to negative values (presented in Table 1-fold change in the situation “*Tnfα* siRNA/scramble”). On the other hand, whereas the expression of anti-apoptotic genes (for instance *Fadd*, *Pak1*, *Tnfrsf17,* and *Traf2*) showed a down-regulation only with noise exposure, the blockage of TNFα by gene silencing led to an up-regulation of the expression of these genes.

Studies reported that in situations with acoustic overstimulation, not only inflammatory responses but also oxidative stress can be involved in the induction of auditory hair cell apoptosis [15,17]. In this sense, it is widely accepted that TNFα is a pro-inflammatory cytokine that, among other properties, can induce auditory hair cell apoptosis through the extrinsic cell death pathway [14]. In addition, studies showed that this cytokine is closely involved with noise-induced hearing loss [7,8]. Research demonstrated that the *Tnfα* levels are increased after noise exposure and that its blockade preserved the hearing [7]. Beyond the TNFα, another well-known pro-inflammatory cytokine, interferon-gamma, (IFNγ), can also be relevant to the hearing loss [18]. Recently, Moon and collaborators [19] demonstrated that, together with TNFα, IFN-γ importantly contributes to cochlear injury in an experimental model (dependent of inflammation) mediated by cisplatin. Although the role of IFNγ was not evaluated here, we agreed that further studies should be developed in order to show the contribution of IFNγ to noise-induced hearing loss.

Corroborating the fact that the inflammatory response and oxidative stress has an important role in the hearing loss, studies demonstrated that the use of glucocorticoids, antioxidants, and JNK inhibitors, which can alter various points in programmed cell death pathways [15], as well as anti-TNFα agents, showed good response in noise-induced hearing loss [8].

Based on these data, strategies to suppress inflammation, especially TNFα, before noise exposure, can be useful to preserve hearing. In agreement with this suggestion, the use of etanercept, which acts by inhibiting TNFα by the decoy receptor, was able to improve cochlear microcirculation and also to avoid hearing loss after loud noise exposure in an animal model [20].

Although the TNFα blockade demonstrated a good perspective in the hearing preservation, to date, there are no studies in cell biology that characterize otoprotection derived from the TNFα blockade using gene therapy. Indeed, small interfering RNA (siRNA) is a new biological tool that induces gene knockdown, allowing the observation of protein function [21].

In relation to the effect of the use of siRNA in the cochlear hair cells gene expression, Mukherjea et al. [22,23,24] observed a silencing rate of 70% for the NADPH oxidase 3 (NOX3) gene after siRNA trans-tympanic injection. In addition, Chen et al. [25] observed a decrease of 30% in the p85α gene expression in hair cells of animals previously submitted to siRNA intra-tympanic injection and after noise exposure. These data corroborated our results, since that the trans-tympanic injection performed in this study was able to reduce the *Tnfα* gene expression to 26.31%, with a silencing rate of 74.1% observed after noise exposure comparing the control. In order to ensure the efficiency of TNFα blockade by gene silencing here, we performed in vitro tests and we found a silencing rate of 95%, demonstrating an adequate response to the selected silencer.

After the observation of the effectiveness of TNFα blockade by gene silencing carried out here, we evaluated the effects of this blockade in the modulation of several genes associated with the TNFα metabolism pathway. Our results showed a significant up-regulation of anti-apoptotic genes, such as *Traf2*, *Tnfrsf17*, and *Pak1*, as well as a downregulation of pro-apoptotic genes, such as *caspase 8* and TNFα family receptors with the death domain, in the cochlea of ears submitted to TNFα blockade by gene silencing.

In relation to the up-regulated genes, TRAF family proteins, especially TNF receptor superfamily member (17*TNFRSF17*/Cd27) and TRAF2 may lead to the activation of transcription factors, such as NFkB and JNK, promoting cell survival and differentiation [26]. In the TNFα context, researchers reported that the deficiency or decrease in TRAF2 expression is related to increased sensitivity of the TNFR1A receptor to cell death TNFα induced [27]. In this study, we found a two-fold change increase in the expression of the *Traf2* gene (Rq: 2.293) and the *Tnfrsf17* gene (Rq: 2.091) in the cochlea submitted to TNFα blockade gene silencing. In a similar way, we also found an eight-fold change increase in the *Pak1* gene (Rq: 8.640) in the cochlea submitted to TNFα blockade gene silencing. 

PAK (or p21 activated kinase) is a Ser/Thr family in which the PAK1 is the main member. PAK1 is able to modulate several pathways involved in the cell proliferation and survival, such as MAPK, AKT (AKT serine/threonine kinase 1), Wnt1/β-catenin, TNFα, BAD (BCL2 associated agonist of cell death), and NF-kB [27]. Pak1 gene expression was already reported in the brain [28] and heart [29]. However, our observation that Pak1 gene expression was up-regulated in the cochlea submitted to TNFα blockade gene silencing not only is a novelty finding but also can putatively suggest that its up-regulation may promote cell survival after the noise-induced hearing loss in our experimental model.

The ability to evade apoptosis has been shown to be induced by a range of alterations including the activation/up-regulation of anti-apoptotic pathways and the inactivation/downregulation of certain apoptotic molecules. The most commonly reported pro-survival strategies are directly antagonizing pro-apoptotic proteins and lead to an increase the expression levels of anti-apoptotic genes [30]. Indeed, here the TNFα blockade by gene silencing led to the downregulation of genes pro-apoptotic, especially related to the extrinsic pathway of cell death [31], such as Caspase 8 can TNFα family receptors with the death domain. These genes are expressed after TNFα action [32]. Therefore, the downregulation of these genes in the cochlea submitted TNFα blockade by gene silencing can be associated with hearing preservation, as found by us.

Regarding the effect of noise exposure in the electrophysiological threshold and amplitude of the waves analyzed in the ears submitted or not to TNFα blockade by gene silencing, we observed significant differences.

As previously mentioned, the auditory brainstem response (ABR), evoked by clicks, is a powerful test used to evaluate the amplitude and latency of the sound-evoked discharge probability of responding fibers, as well as the number of fibers responding synchronously. In terms of the ABR click test, it is based on a stimulus with a constant spectral density at all frequencies up to the sampling limit, exciting the auditory system strongly at a wide range of frequencies within a very short period of time. In order to achieve a strong response, the onset of the click needs to be as steep as possible [33]. 

The wave I in rodents is the same in humans and represents the cochlear nerve, whereas the wave IV in rodents corresponds to wave V in humans. In addition, the wave II represents the largest and stable wave used to evaluate clinical parameters [34,35]. In this sense, Kujawa and Liberman [36] related decreased wave I amplitude with persistent loss of the synaptic connections between the terminal nerve fibers and inner hair cells, which characterizes synaptopathy. Stamper and Johnson [37] observed a decrease in the wave I amplitude in ABR click in humans after noise exposure, similar to the animal model. So, our results showed that the mean electrophysiological threshold shift in the silenced ears was 49.5 dBSPL +/−10.5 dBSPL, which was statistically higher than the noised ears (*p* < 0.001) and also that at 80 dBSPL wave I amplitude was statistically higher in silenced ears compared to control ears.

Changes in the wave amplitude and morphology represent auditory responses of the brainstem ascending auditory pathway affected by the cochlear lesion. Peripheral lesions are reflected in the displacement of the auditory thresholds and weaker responses to sound at various levels of the central auditory system, from the cochlear nucleus to the auditory cortex [38]. Corroborating these pieces of information, Popelar et al. [39] demonstrated that the amplitudes evaluated by ABR in rats were significantly altered after noise exposure with the maximal suppression immediately after exposure (measured one-day post-exposure). Similarly to described above, we also found significant alterations. In 80 dBSPL, the wave I amplitude was statistically higher, while at 90 dBSPL, all wave amplitudes were statistically higher in the silenced ears when compared with the non-silenced ears.

These results showed that the administration of *Tnfα* siRNA in one ear did not influence in the electrophysiological parameters found in the control ear. Corroborating this fact, Oishi et al. [40] administered an siRNA for Nox3 in one ear of mice and the other was used as control. The authors evaluated the protein NOX3 fluorescence in the outer hair cells and reported a significant reduction of the fluorescence at 72 h after administration in the ear treated with siRNA as compared to the control ear.

Despite our results demonstrating that the TNFα blockade by gene silencing was able to modulate the expression of several genes involved in the TNFα metabolism pathway and also to prevent the noise-exposure hearing loss, we understand that a limitation of the study is the lack of results showing the quantity of protein “TNFα“ in the ears submitted or not to gene silencing and noise exposure. However, we believe that our results can motivate and guide other authors to develop further studies in order to increase the knowledge regarding the effects of TNFα blockade by gene silencing in the context of noise-exposure hearing loss.

Taken together, our results reinforce the previous observation that TNFα has an essential role in noise-induced hearing loss, especially by inducing cochlear cell apoptosis through the extrinsic pathway activation, which led to the alterations not only in the electrophysiological threshold shift but also in the wave amplitude related to the synaptopathy. In addition, for the first time, we were able to demonstrate that the TNFα blockade by gene silence was useful to suppress the deleterious actions of this pro-inflammatory cytokine in our experimental model.

## 4. Material and Methods

### 4.1. Materials

*Tnfα* siRNA (Thermo Fisher Scientific Cat N^o^: AM 16830); scramble siRNA (Thermo Fisher Scientific Cat N^o^: AM16230); Pure Link^®^ RNA Mini Kit (Thermo Fisher Scientific Cat. N^o^: 12183018A); Pure Link^®^ DNase for on-column (Thermo Fisher Scientific, Cat. N^o^:12185010); SuperScript^®^ VILO™ Master Mix (Thermo Fisher Scientific cat. N^o^: 11755050); TaqMan^®^ Array Fast Plates (Thermo Fisher Scientific Cat N^o^: 1605216-001); RNAlater (Qiagen, Valencia, CA, USA); and TRIzol^®^ Reagent (Thermo Fisher Scientific, Cat. N^o^: 15596018) were used. Dharmacon™ Accell™ siRNA Delivery (Horyzon Discovery, UK) was used to carried out the in vitro assays. TNFα antibody fluorescent tagged (dylight 488) were purchased from Santa Cruz Biotechnology (Santa Cruz, CA). C6 cell line: Brain fibroblast from *Rattus norvegicus*. Code: 0057 was purchased from the Rio de Janeiro cell bank (BCRJ).

### 4.2. Animal Preparation

Young male Wistar rats (*n* = 20, weight = 50 g, aged between 4–5 weeks) with normal Preyer’s reflex and 50 dBSPL threshold shift in auditory brainstem responses (ABR) to clicks were used. All procedures were approved by the University of São Paulo/Brazil ethics committee on Laboratory Animal Medicine, in accordance with the Guide for the Care and Use of Laboratory Animals (National Council for Control of Animal Experimentation/ Brazil), license *n*. 146/15.

The animal ears were randomly assigned by www.randomization.com to receive trans-tympanic siRNA application against TNFα (siRNA TNFα group) or scramble siRNA (negative control).

### 4.3. Auditory Brainstem Response

The threshold shifts were tested by click-evoked ABR under isoflurane anesthesia (1.8%) in a sound-attenuating chamber, using a TDT System 3 workstation (Tucker Davis Technologies, Alachua FL, USA), before gene silencing and one day after noise exposure, in order to determine the magnification of the noise-induced hearing loss under experimental noise conditions. 

The acoustic stimuli were presented at decreasing intensities, stepwise, from 90 dBSPL to 10 dBSPL, using an MF-1 magneto electrostatic speaker placed 5 cm from the animal’s ear, with occlusion of the other ear. The ABR measurements were carried out using methods and equipment reported previously [34,35,41,42]. The threshold shifts at each intensity were verified at least twice and considered when wave II could no longer be visualized [34,35].

### 4.4. Trans-Tympanic Injection

Rats were intraperitoneally anesthetized with a mixture of xylazine 10 mg/kg and ketamine 50 mg/kg. The external auditory canal and the tympanic membrane were examined under a surgical microscope and after an additional incision in the *pars tensa*, to allow air evacuation during the injection. Ten microliters (10 µL) from a siRNA solution (containing 625 pmoL in water RNAse free) were gently and directly injected to the inner ear without any packaging using a glass micro-syringe connected to a 34 G catheter trans-tympanically [43,44,45]. As a control, the ear that received the scramble responded in the same way to the ear only exposed to noise (data not shown).

### 4.5. Loud Noise Exposure

The animals were placed awake in a wire cage inside a box with acoustic insulation, exhaustion, and internal air circulation, measuring 760 × 485 × 705 mm, associated with an EP 125 Audio Signal generator with free access to water and food. The applied stimulus was high frequency white noise (12,000 Hz at 20,000 Hz), at a sound intensity of 120 dBSPL for three hours, produced by a sound generator and reproduced through loudspeakers positioned in the upper part of the box above the animal cage (Insight© Ribeirão Preto/SP). The noise level across the box was constant, measured through an AK 824 digital decibel meter. This noise level was chosen because it is capable of inducing permanent hearing loss [46,47,48].

### 4.6. RNA Isolation and REAL-TIME -qPCR

One day after noise exposure, the animals were decapitated, and their bullae was rapidly removed from their skull and placed into a cold RNA stabilizing reagent (RNAlater). After that, the cochlea was isolated and transferred to RNase-free PCR tube with 1 mL cold TRIzol^®^ Reagent and homogenized using a rotor station. Subsequently, 0.2 mL of chloroform was added to the tube and shaken vigorously by hand for 15 s and then centrifuged at 12,000× *g* for 15 min at 4 °C. Then, 400 μL of the colorless phase containing RNA was transferred, followed by the addition of an equal volume of 70% ethanol. About 700 μL of this solution was transferred to a collection tube of PureLink^®^ RNA Mini Kit. DNase treatment was carried out using the PureLink^®^ DNase treatment protocol in the column described in detail at Bio-protocol [49].

The RNA pellet was resuspended in nuclease-free water and the RNA levels were determined using optical density readings corresponding to wavelengths 260/280 nm using a spectrophotometer (NanoDrop^TM^ 2000/2000c from Thermo Fisher Scientific). The total RNA was converted to cDNA using the SuperScript^®^ VILO™ Master Mix and applied to real-time qRT-PCR for *Tnf*α using TaqMan^®^ Array Fast Plates according to the manufacturer’s protocol. The TaqMan Gene Expression Assay included 96-well plates with pre-designed assay based on the TNFα metabolic pathway.

### 4.7. In Vitro Silencing Analysis with High-Content Screening

Glial fibroblast cells from *Rattus novergicus* (F98) purchased from a Rio de Janeiro cell bank (BCRJ) were used. The cells were cultured in F-12K medium suitable for high-content screening. Two independent experiment sessions were carried out, both in triplicate, with *Tnfα* siRNA as the gene of interest, using the commercial kit Dharmacon™ Accell™ siRNA Delivery (Horyzon Discovery, Cambridge, UK). Nine sites per well and three wells per treatment were analyzed. The presence of the TNFα protein was determined using ImageXpress and TNFα positive cells were determined using the MetaXpress software.

The in vitro silencing rate was calculated by the formula: Intensity of green labelling in cells without *Tnfα* siRNA treatment.
Intensity of green labelling in cells with *Tnfα* siRNA treatment

### 4.8. Data Analysis

The comparative ΔΔCT method was used to calculate the relative gene expression (Rq), using the formula 2^−ΔΔ*C*T^ which corresponds to how many times the target gene was over- or under-expressed, when comparing the test samples to the control samples [50].

The expression level of each gene was normalized to a mean level of the reference genes (Glyceraldehyde 3 phosphate dehydrogenase (*Gadph),* hydroxymethylbilane synthase *(Hmbs*), actin beta *(Actb)* and *18S*), in order to generate the ΔCt. The ΔΔCt was then calculated using the following formula: ΔCt (noise group) −ΔCt (noise group), followed by the application of the 2^−ΔΔCT^ formula.

To determine the fold change, the formula −1/Rq was applied for values below 1 [50,51]. The fold change was then considered biologically significant when over or equal (≥) to 1.2657 (95% CI = 1.095−1.4627) for increased expression, and less or equal (≤) to −1.2657 (95% CI = 1.095−1.4627) for decreased expression [52]. The percentage of the in vivo knockdown or silencing rate was calculated by the formula 100 (−2^−ΔΔ*C*T^ × 100), according to the description in the link:

https://www.thermofisher.com/br/en/home/references/ambion-tech-support/rnai-sirna/tech-notes/understanding-calculations-for-sirna-data.html.

Data was stored in the BioSigRZ System III software (Tucker-Davis Technologies, Alachua, FL, USA) for offline analysis. Means, standard deviations, and adjusted *p*-values were obtained using Student’s *t*-test with Bonferroni’s post-hoc test comparing the values of ears that received *Tnfα* siRNA or noise. Student’s *t*-test was also used in vitro analysis [53]. The significance level was set at 5%.

## Figures and Tables

**Figure 1 ijms-21-02692-f001:**
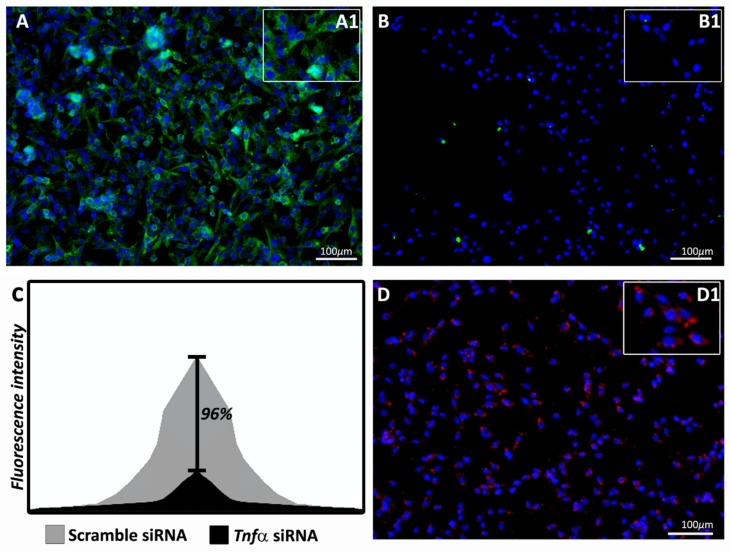
Fluorescence analysis of tumor necrosis factor α (TNFα) (green-labeled) in glial fibroblast cells treated with scramble small interfering RNA (siRNA) (negative control-A) or *Tnfα* siRNA (**B**). Shown in (**C**) is the silencing rate of the fluorescence intensity between the scramble siRNA and *Tnfα* siRNA (black bar, 96%). In (**D**) is the positive control of the delivery agent (red-labeled). The fluorescence of nine sites per well, with a total of three wells per treatment, was analyzed by MetaXpress software. Statistical differences in the values of TNFα labeling (**A** and **B**) were obtained using Student’s *t*-test, with the Bonferroni post-hoc test, at a significance level of 5% (*p* < 0 05).

**Figure 2 ijms-21-02692-f002:**
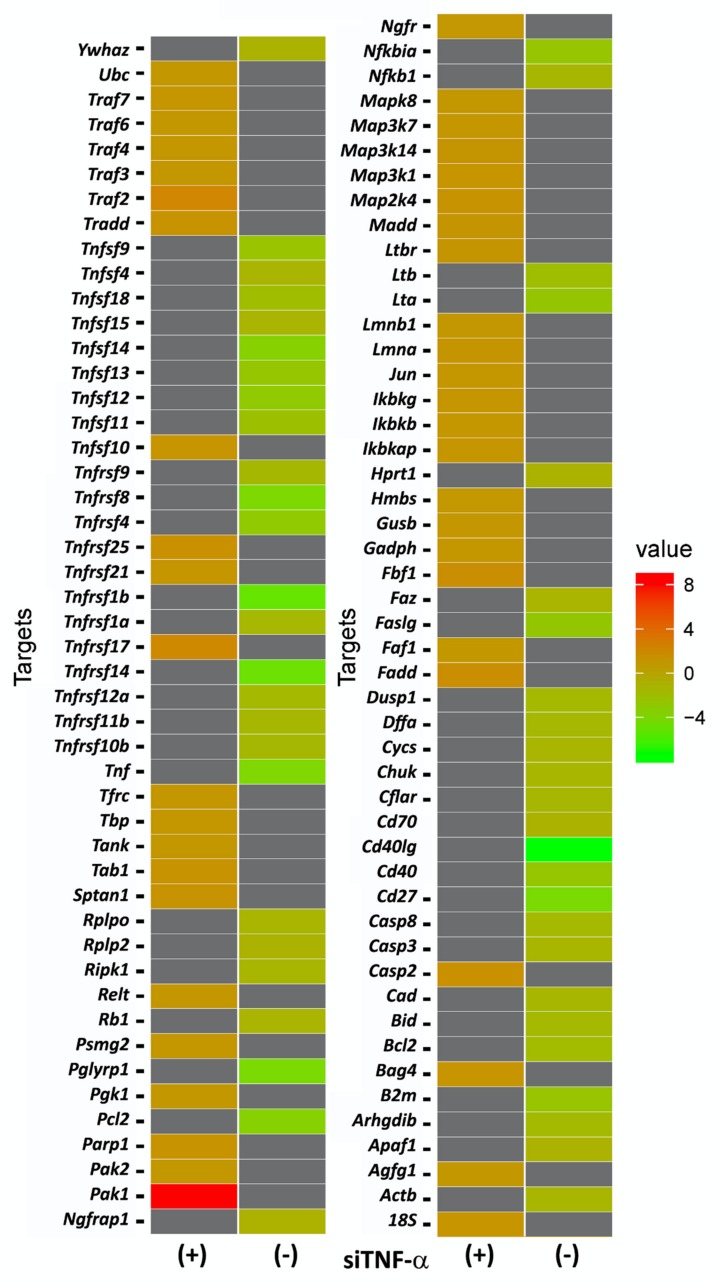
A heat map showing the comparisons of differential gene expression of the TNFα metabolic pathway in the cochleae of rats previously submitted (*n* = 20) or not submitted (*n* = 20) to *Tnfα* siRNA administration and after noise exposure. According to the fold-change found by the qRT-PCR analysis, the “red” color is used to indicate the highest gene expression (up-regulated genes), whereas the “green” color is used to indicate the lowest gene expression (down-regulated genes). In the “gray” color are presented the gene expression values (or transcript values) that were poorly evaluated due to insufficient resolution or image noise, which, in a general way, are named as “missing values”.

**Figure 3 ijms-21-02692-f003:**
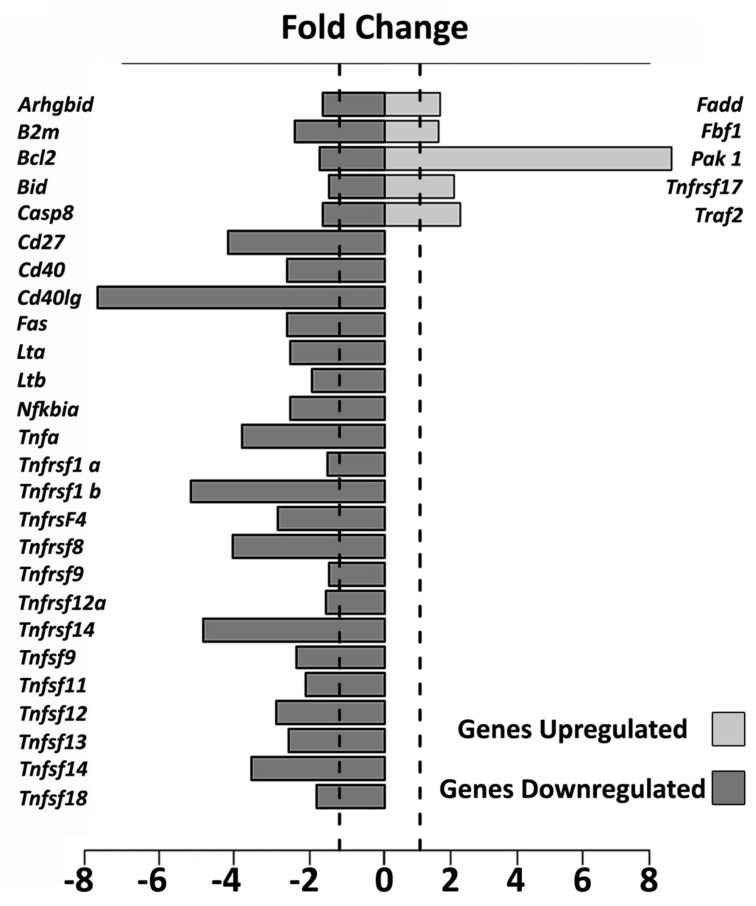
Fold change of the main apoptotic genes that were up and down-regulated in the cochleae of rats previously submitted (S, *n* = 20) or not submitted (NS, scramble, *n* = 20) to TNFα blockade by gene silencing before and after noise exposure.

**Figure 4 ijms-21-02692-f004:**
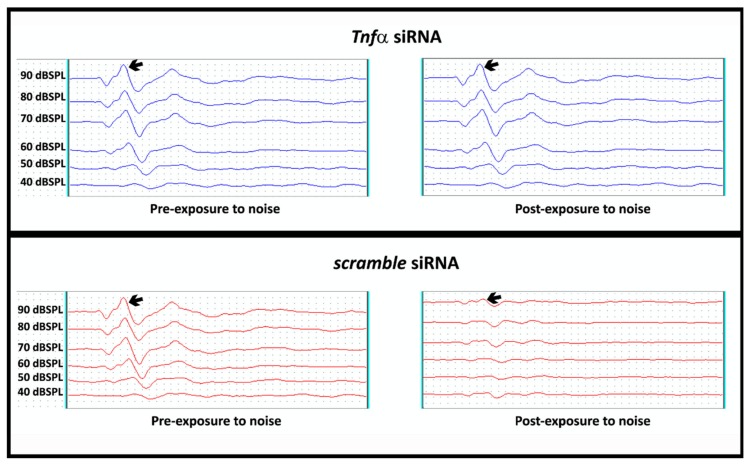
Representative analysis of the electrophysiological threshold shift (in dBSPL) in the cochleae of rats previously submitted (S, *n* = 20) or not submitted (NS, scramble, *n* = 20) to TNFα blockade by gene silencing before and after noise exposure. Narrow represents wave II, which was used to determine the threshold shift.

**Figure 5 ijms-21-02692-f005:**
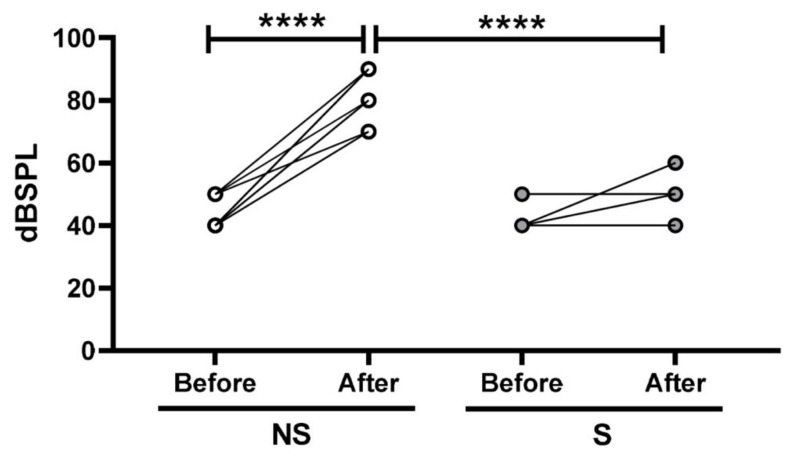
Analysis of the electrophysiological threshold shift (in dBSPL) in the cochleae of rats previously submitted (S, *n* = 20) or not submitted (NS, scramble, *n* = 20) to TNFα blockade by gene silencing before and after noise exposure. Differences in the electrophysiological threshold shift were evaluated by Student’s *t*-test with Bonferroni’s post-hoc test. The significance level was set as 5%. **** *p* < 0.0001.

**Figure 6 ijms-21-02692-f006:**
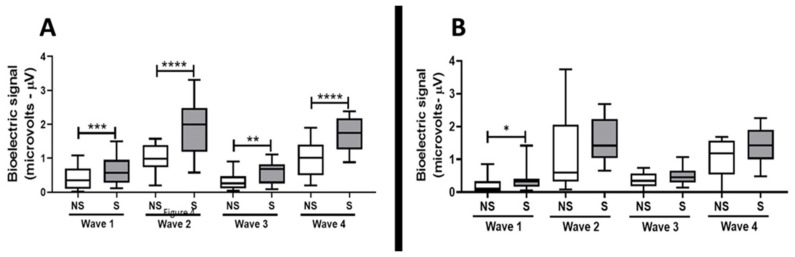
Auditory brainstem response (ABR), evaluated by waves amplitude I, II, III, and IV (represented by bioelectric signals set at microvolts, μV) at 90 dBSPL (**A**) and at 80 dBSPL (**B**) in ears previously submitted (S, *n* = 20) or not submitted (NS, scramble, *n* = 20) to TNFα blockade by gene silencing and after to noise exposure. Differences in the wave’s amplitude were evaluated by Student’s *t*-test with Bonferroni´s post-hoc test. The significance level was set as 5%. * *p* < 0.05, ** *p* < 0.01, *** *p* < 0.001, **** *p* < 0.0001.

**Table 1 ijms-21-02692-t001:** Description of the grouping of the genes analyzed in the TNFα metabolic pathway performed by qRT-PCR analysis in the cochleae of rats previously submitted (*n* = 20) or not submitted (*n* = 20) to *Tnfα* siRNA administration and after noise exposure, in their respective families. In addition, the fold changed values of siRNA *Tnfα*/scramble, as well as of noise/no noise is shown. Higher and lower gene expressions in the fold changes of siRNA *Tnfα*/scramble were determined by the cut-off defined at 1.2657. Samples were normalized with the *Gadph, Hmbs, Actb,* and *18S* genes. Abbreviation: B-cell CLL/lymphoma 2 (Bcl), Mitogen-activated protein kinase (Mapk), TNF receptor associated factor (Traf), Nuclear factor kappa-light-chain-enhancer of activated B cells ( NfkB ).

	Fold Change *Tnfα siRNA*/*Scramble*	Fold Change Noise/No Noise		Fold Chang e*Tnfα siRNA*/*Scramble*	Fold Change Noise/No Noise
**Bcl Family Intrinsic pathway**			**Caspase Famil**y		
*Bcl2*	−1.73	−1.64	*Casp 3*	−1.26	1.24
*BID*	−1.57	1.22	*Casp 8*	−1.64	1.30
*Cytochrome c*	−1.18	−1.36			
*Apaf 1*	−1.09	−1.19	**TNF ligand Famil**y		
			*Cd40L*	−7.63	−1.75
**TNF receptors Famil**y			*Tnfa*	−3.85	2.065
*Cd27*	−4.18	−1.46	*Tnfsf4*	−1.18	−1.23
*Cd40*	−2.61	1.25	*Tnfsf9*	−2.36	1.03
*Lta*	−2.54	1.17	*Tnfsf10*	1.23	−1.89
*Ltb*	−1.96	−1.56	*Tnfsf11*	−2.11	−1.03
*Ltbr*	1.19	−1.08	*Tnfsf12*	−2.88	1.0
*Tnfrsf1A*	−1.52	1.18	*Tnfsf13*	−2.5	−1.31
*Tnfrsf1B*	−5.18	−1.15	*Tnfsf14*	−3.5	1.549
*Tnfrsf4*	−2.84	1.09	*Tnfsf15*	−1.21	−1.38
*Tnfrsf8*	−4.07	1.18	*Tnfsf18*	−1.8	−1.70
*Tnfrsf9*	−1.49	−1.15	*FasL*	−1.15	−1.24
*Tnfrsf10B*	−1.48	−1.05			
*Tnfrsf11B*	−1.39	−1.09	**Traf Family**		
*Tnfrsf12A*	−1.56	1.1	*Traf2*	2.293	−1.28
*Tnfrsf14*	−4.84	−1.36	*Traf3*	1.03	−1.29
*Tnfrsf 17*	2.09	−5.02	*Traf4*	1.048	−1.2
*Tnfrsf21*	1.16	−1.01	*Traf6*	−1.03	−1.19
*Tnfrsf25*	1.47	−1.09	*Traf7*	1.142	−1.07
*Fas*	−2.68	1.04			
*Fbf1*	1.61	−1.17			
**Death domain Family**			**MapK Family**		
*Fadd*	1.65	−1.43	*Map3K1*	1.253	4.52
*Tradd*	1.35	−1.53	*Map2K4*	1.263	−1.23
			*Map3K7*	1.011	4.52
**NfkB Family**			*Mapk8*	−1.07	−1.38
*NfkB1*	−1.3	−1.31	*Map3k14*	1.18	−5.02
*NfkBia*	−2.51	−2.60	*Jun*	1.07	1.07
*Ikbkg*	1.11	−1.10			
*Ikbkb*	1.02	−1.10	**Anti-apoptotic genes **		
*Ikbkap*	1.158	−1.19	*Pak1*	8.640	−1.05
			*Tradd*	1.35	−1.53

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
