# Peer review of "New Insights on the Effect of TNF Alpha Blockade by Gene Silencing in Noise-Induced Hearing Loss"

_ijms, 2020, doi:10.3390/ijms21082692_

Round 1
Reviewer 1 Report
This study reported the effect of silencing TNF-α on noise-induced hearing loss. TNF-α siRNA was pretreated to one side ear of Wistar rat before white noise exposure. The ABR results after noise exposure showed that the electrophysiological threshold and waves amplitude were prevented. In addition, the qRT-PCR results showed the increase of anti-apoptotic genes expression and decrease of pro-apoptotic genes expression after TNF-α blocked. In conclusion, the blockade of TNF-α prevented the noise induced hearing loss. This could be a new insight on protecting the noise induced hearing loss, but the quality of the whole manuscript and figures is very terrible. The results provided in the manuscript cannot support the conclusion very well. I don’t think this manuscript is suitable to publish on this journal. I also have some suggestions to make the manuscript better.
Major comments:
- In the results section, you’d better give a subheading for each section. And the description of each result is not detailed enough.
- Figure 1 is not very clear; you should provide the representative figure with high resolution. Also, there is no statistical chart for figure 1.
- The resolution of figure 2 is really poor to recognized any information, also more detail description should be provided in figure legend.
- The figure legends are roughly described, you’ better give more detail information, such as the sample number, the information of the frequencies you measured etc.
- Why the author chose to analysis wave Ⅱ, Ⅲ and Ⅳ amplitude, more detail information need to provide.
- There are no experiments to verify the silence efficiency of TNF-α siRNA.
- The frequencies measured in the ABR experiment need to provide
- Where is the qRT-PCR results?
- Why you chose the Glial fibroblast cells for the in vitro silencing analysis?
- The ABR measurement need to performed after TNF-α siRNA treatment to make sure no obvious hearing loss because of the surgery.
Minor comments
- Please keep the TNF- α written correctly, for example, in line 68, the “α” was written to “a”, please check the writing!
- Line 78, what’s the age of the rat.
- Please confirm the format of the reference number, in line 51, the format is “[9,10,11]”, while in line 102, “[18-20]”.
- In line 95, the font size of “15,16” is obviously larger than others!
- In Figure4, what the vertical axis represented? It’s confusing.
Reviewer 2 Report
TNF-alpha has an important role in noise-induced hearing loss. In this manuscript, the authors have blocked TNF-alpha using siRNA. Although it already has been described the inhibition of TNF using drugs, this is the first work describing the use of siRNA. Authors show that the blockade of TNF alpha increases anti-apoptotic genes expression and decreased pro-apoptotic genes expression. Moreover, TNF-alpha inhibition reduces the effect of the noise-induced hearing loss in rats after transtympanic siRNA application.
In order to improve this manuscript, I think some revisions and explanations need to be provided.
Line 42 maybe the following reference should be added Landegger et al 2019. Cytokine Levels in Inner Ear Fluid of Young and Aged Mice as Molecular Biomarkers of Noise-Induced Hearing Loss
Line 62 reference 13 the size of the number is too high
Line 166 in order to show the silencer level, it would be very informative to include a histogram showing the qPCR values of TNF alpha before and after siRNA administration.
Figure 2 has not enough quality. It is impossible to see the genes name.
Figure 2. What does “missing value” mean? It is not clear if the author didn't obtain value for these genes or these values were too low.. Clarify this point, please.
table1 header. Description of total genes….I think is not correct because authors don't show any description f the gene, then it should be rephrased. The number of cochleae used should be added.
Figures 3 and 4 .could you shown the number of animals analyzed, please?
Line 221, It is not clear what “positively-regulated” means. Do you what to say up-regulated?
At the end of the article, there is a table. First of all the table header is not in English. This reviewer doesn't know which is the number of this table, because I can not find any reference across the text.
It would be interesting to compare histological preparations of rats' inner ear after siRNA administration and noise exposition with non-administration and noise-exposed rats.
Round 2
Reviewer 1 Report
This study reported the effect of silencing TNF-αon noise-induced hearing loss. TNF-α siRNA was pretreated to one side ear of Wistar rat before white noise exposure. The ABR results after noise exposure showed that the electrophysiological threshold and waves amplitude were prevented. In addition, the qRT-PCR results showed the increase of anti-apoptotic genes expression and decrease of pro-apoptotic genes expression after TNF-α blocked. In conclusion, the blockade of TNF-α prevented the noise induced hearing loss. Even this study had been revised, the problems we asked last time wasn’t solved at all. The author did not perform any of the additional experiment that I mentioned with very ridiculous excuse, such as can’t buy the antibody. The quality of the whole manuscript still roughly and results cannot support the conclusion very well. I don’t think this manuscript is suitable to publish on this journal. I still insist that my previous concerns need to be addressed to make the manuscript better.
Major comments:
- The representative image in figure1 has been changed, however the statistical information still didn’t be provided. This is not suitable to descript the statistical information instead of statistical information.
- Even the author changed the image in figure 2, the font size and color of the genes name is not suitable, we suggest using Photoshop software to improve image quality.
- Without any silence efficiency experiment of TNF-α, the reader will not believe your results in this study. This is necessary to demonstrate TNF-α siRNA worked well.
- The qRT-PCR results is really important mechanism study, the results should be placed in mainly figures. Also, the qRT-PCR results should be represented as bar graphs instead of supplementary tables.
- There are many results the author observed but not provided in this study, for example the ABR data after surgery and siRNA silencing efficiency.
- Only 4 figures the author provided in this study. This is not enough for publication.
- The author only test mRNA of TNF- α metabolic pathway, however, western blot really need to double confirm these proteins expression.
Minor comments
- Please keep the TNF- α gene written consistently, for example, TNF-α and Tnf-α
Reviewer 2 Report
I have seen another mistakes
line 208 and 211 "painel" need to be change to panel.Please, ckeck the spelling again across the manuscript.
Authors have clarify all the question.
Author Response
Reviewer #2
Comments and Suggestions for Authors
I have seen another mistakes.
Line 208 and 211 “painel” need to be change to panel. Please, check the spelling again across the manuscript.
Authors have clarify all the question.
Author´s response: First of all, on behalf of all authors, I would like to thank you very much for the excellent revision of our manuscript, which helps us to improve it.
As required, the word “painel” was changed to “panel”. In addition, the spelling was also checked.
Round 3
Reviewer 1 Report
This study reported the effect of silencing TNF-α on noise-induced hearing loss. TNF-α siRNA was pretreated to one side ear of Wistar rat before white noise exposure. The ABR results after noise exposure showed that the electrophysiological threshold and wave amplitude were prevented. Also, the qRT-PCR results showed the increase of anti-apoptotic genes expression and decrease of pro-apoptotic genes expression after TNF-α blocked. In conclusion, the blockade of TNF-α prevented the noise-induced hearing loss. This study had been revised, the problems we asked last time have been solved. I think this manuscript can be published since enough supplementary data provided. In addition, I still have two minor suggestions to make the manuscript better.
Minor comments:
- You’d better give the statistical chart in Figure1 instead of the statistical information because the statistical chart is more intuitive.
- Please keep the gene name written consistently: italicized and first letter upper case all the rest lower case. Especially in figure 2, figure S1 and figure S2. Please confirm other gene names in the whole manuscript and figures.
Author Response
Responses to Reviewer comments
Reviewer #1
Comments and Suggestions for Authors
This study reported the effect of silencing TNF-α on noise-induced hearing loss. TNF-α siRNA was pretreated to one side ear of Wistar rat before white noise exposure. The ABR results after noise exposure showed that the electrophysiological threshold and wave amplitude were prevented. Also, the qRT-PCR results showed the increase of anti-apoptotic genes expression and decrease of pro-apoptotic genes expression after TNF-α blocked. In conclusion, the blockade of TNF-α prevented the noise-induced hearing loss. This study had been revised, the problems we asked last time have been solved. I think this manuscript can be published since enough supplementary data provided. In addition, I still have two minor suggestions to make the manuscript better.
Minor comments:
Reviewer (suggestion 1). You’d better give the statistical chart in Figure1 instead of the statistical information because the statistical chart is more intuitive.
Author´s response: First of all, on behalf of all authors, I would like to thank you very much for the excellent revision and suggestion regarding our manuscript, which helps us to improve it.
Author´s response: First of all, on behalf of all authors, I would like to thank you very much for the excellent revision and suggestion regarding our manuscript, which helps us to improve it.
As required, the statistical chart in Figure 1 was added (Figure 1C).
Reviewer (suggestion 2). Please keep the gene name written consistently: italicized and first letter upper case all the rest lower case. Especially in figure 2, figure S1 and figure S2. Please confirm other gene names in the whole manuscript and figures.
Author´s response: Thank you very much for the recommendation and I would like to inform that all genes names were checked and corrected as required.